# Evaluation by Flow Cytometry of *Escherichia coli* Viability in Lettuce after Disinfection

**DOI:** 10.3390/antibiotics9010014

**Published:** 2019-12-31

**Authors:** Pilar Teixeira, Bruna Fernandes, Ana Margarida Silva, Nicolina Dias, Joana Azeredo

**Affiliations:** 1LIBRO—Laboratório de Investigação em Biofilmes Rosário Oliveira, Centre of Biological Engineering, University of Minho, 4710-057 Braga, Portugal; brunacfernandes.09@gmail.com (B.F.); 26anamar@sapo.pt (A.M.S.); jazeredo@deb.uminho.pt (J.A.); 2Centre of Biological Engineering, University of Minho, 4710-057 Braga, Portugal; nidias@deb.uminho.pt

**Keywords:** flow cytometry, viable but nonculturable cells (VBNC), lettuce, chemical disinfection

## Abstract

Foodborne outbreaks due to the consumption of ready-to-eat vegetables have increased worldwide, with *Escherichia coli* (*E. coli*) being one of the main sources responsible. Viable but nonculturable bacteria (VBNC) retain virulence even after some disinfection procedures and constitute a huge problem to public health due to their non-detectability through conventional microbiological techniques. Flow cytometry (FCM) is a promising tool in food microbiology as it enables the distinction of the different physiological states of bacteria after disinfection procedures within a short time. In this study, samples of lettuce inoculated with *E. coli* were subject to disinfection with sodium hypochlorite at free chlorine concentrations of 5, 10, 25, 50, and 100 mg·L^−1^ or with 35% peracetic acid at concentrations of 5, 10, 25, and 50 mg·L^−1^. The efficiency of these disinfectants on the viability of *E. coli* in lettuce was evaluated by flow cytometry with LIVE/DEAD stains. Results from this study suggest that FCM can effectively monitor cell viability. However, peracetic acid is more effective than sodium hypochlorite as, at half the concentration, it is enough to kill 100% of bacteria and always induces a lower percentage of VBNC. Finally, we can conclude that the recommended levels of chemical disinfectants for fresh fruit and vegetables are adequate when applied in lettuce. More importantly, it is possible to ensure that all cells of *E. coli* are dead and that there are no VBNC cells even with lower concentrations of those chemicals. These results can serve as guidance for lettuce disinfection, improving quality and the safety of consumption.

## 1. Introduction

The importance of a healthy lifestyle triggered changes in the diet of populations, namely, a higher consumption of fruit and vegetables. These products can be consumed raw or minimally processed but to prepare meals quickly and to retain freshness, vegetables are usually consumed pre-packed. Leafy greens consumed raw as salads were considered the highest priority group in terms of fresh produce safety [1]. Just in the United States, 606 leafy vegetable-associated outbreaks with 20,003 associated illnesses, 1030 hospitalizations, and 19 deaths were reported to the Center for Disease Control and Prevention between 1973 and 2012. Of the 162 (27%) outbreaks due to a single leafy vegetable, most were attributed to lettuce (120, 74%) [2] and mostly related to *Escherichia coli* O157:H7 or *Salmonella* spp. [3,4]. In 2018, in the United States, the *E. coli* outbreak stemming from romaine lettuce had spread to 36 states and infected at least 210 people, of whom 96 were hospitalized and resulted in five deaths [5]. In fact, the introduction of these pathogens throughout the production chain is straightforward, namely through contact with improperly treated manure compost, contaminated irrigation water, the presence of wild and domestic animals, and unclean containers and tools used in harvesting [6]. Contamination may also occur during postharvest handling, processing, and distribution [6]. Unfortunately, there are no strategies able to achieve the complete elimination of hazardous microorganisms on produce without compromising their quality and without reducing shelf life [7]. Due to the low cost associated with its use and its ease of application, hypochlorite is the most widely used disinfectant in the fresh-cut industry. Moreover, chlorine is highly efficient causing cell death in pathogens such as *E. coli*. It attacks the bacterial cell membrane, consequently leading to decreased cell permeability and interruption of many other functions [8]. As a drawback, a reaction of chlorine with trace amounts of organic material on fresh produce can occur to form various organochlorine compounds, which are carcinogenic [8]. Moreover, during produce washing, the level of free chlorine in the wash water can change rapidly because chlorine efficacy is affected by the organic loads in the washing solution. Freshly cut lettuce can release abundant amounts of vegetable latex into the washing solution and quickly deplete free chlorine [9]. Thus, instead of specifying a target prewash chlorine level, industry guidelines have suggested that wash water management focuses on maintaining a sufficient level of residual free chlorine [10]. Several values have been suggested and, although these values can be followed as a guide, the determination of the minimum effective chlorine level may be addressed for each case considering the factors that may affect the antimicrobial efficacy of chlorine [11]. To overcome these difficulties, other agents like chlorine dioxide, ozone, organic acids, peracetic acid, and hydrogen peroxide have gained interest as alternative sanitizing agents [12]. Peracetic acid (PAA) has been studied as a disinfectant for washing a variety of fruit and vegetables because it is an effective bactericide, sporicide, fungicide, and virucide that acts across cellular membranes, oxidizing molecules, and destroying essential enzymatic systems [13]. It has stability in the presence of organic matter and it does not produce harmful disinfection by-products [14]. The only disadvantage of PAA is the higher cost involved when compared to chlorine. Nonetheless, disinfectants are often ineffective as microbial reductions above two log units are not achieved, probably because microorganisms are protected by their location in the plant tissue (internalized through the stomata, in cracks, crevices, and cut surfaces) and by biofilms.

Another critical problem is the existence of foodborne bacteria in a state of latency induced by stressful conditions, such as low temperature, high osmolarity, and nutrient starvation. These bacteria called “viable but nonculturable (VBNC) cells” have a discrete metabolic activity, will not grow in standard solid culture media, and are not able to replicate and thus easily escape from conventional detection methods as colony forming units (CFU). Under appropriate conditions of nutrition and temperature, VBNC bacteria can be cultured again and keep their virulence. VBNC cells can hardly be distinguished from live and dead cells but compared with dead cells, VBNC cells have an intact membrane, maintain gene expression, and avoid cytoplasmic leakage. In contrast, dead cells have a damaged membrane and are metabolically inactive [15].

In food, the presence of these damaged bacteria can be critical due to their potential excretion of toxic or food spoiling metabolites [16,17]. As with other bacteria, to resist adverse environments, *E. coli* may gradually enter a VBNC state and thus have been implicated in several diseases [18]. As these cells cannot be detected using traditional plate counting methods, they can cause a false-negative result, which poses a potential threat to food safety and human health.

Staining is a basic step for determining viability and is the most useful method for detecting VBNC. This technique is based on the integrity of the cell membrane and considers the cells with a damaged membrane to be dead. VBNC and viable cells are cells with an intact membrane [18]. Therefore, with the help of flow cytometry or fluorescent microscopy, viable cells can be distinguished from dead cells based on color. Enumerating bacteria, FCM has the advantage of shortly indicating the viability of the samples subject to disinfectants, in contrast to the classical method based on culture which takes at least 24 h. The VBNC state has to be confirmed by both culture plating and viability staining experiments in parallel.

In this context, the main aim of the present study was to make a thorough analysis of the efficacy of chemical disinfectants by assessing *E. coli* viability in lettuce (*Lactuca sativa L*.) with flow cytometry.

## 2. Results and Discussion

The effect of different concentrations of two commonly used disinfectants, sodium hypochlorite and peracetic acid on *E. coli* viability was assessed on lettuce through flow cytometry using Live/Dead staining. SYTO-BC (green fluorescent nucleic acid stain) and propidium iodide (PI) were used as fluorescent probes to determine the proportions of the different viability states of bacteria in cell suspensions. PI is excluded from cells with structurally intact cytoplasmic membranes because of the size and charge of the molecule [19]. SYTO dye, in contrast to PI, is a cell-permeant molecule entering in both live (viable) and dead (non-viable) cells. Accordingly, viable and dead cells were defined by the staining profile of SYTO+PI− (green) and SYTO−PI+ (red), respectively [20]. Intermediate cellular states characterized by the degree of damage afflicted on the bacterial membrane presented staining characteristics of both live (SYTO+) and dead (PI+) cells. This double-positive staining of the bacteria is defined by several reports as compromised or in an intermediary stage between live and dead [20,21,22,23]. These damaged cells emit both green and red, according to the amount of propidium iodide that can penetrate the cell, depending on the extent of the damage.

For washing fresh leafy vegetables and herbs the recommended concentrations of free chlorine range from 50 to 100 ppm [7]. Some factors may affect the antimicrobial efficacy of disinfectants. However, in this study, no attempt was made to evaluate the effect of those factors and only different concentrations of disinfectants were assayed. Accordingly, we tested concentrations of chlorine up to the maximum allowable limit (5, 10, 25, 50, and 100 mg·L^−1^). Figure 1A represents an example of FCM analysis of cell viability. Four cell populations can be distinguished: cells with an intact membrane (C—viable), cells with a damaged membrane (A—dead), and cells with a compromised membrane (B—compromised). The fourth population includes the unstained debris located in the lower-left quadrant. Disinfectants can cause cell lysis and, consequently, the loss of their nucleic acids, which causes them to remain unstained, thereby increasing the unstained population. Moreover, cells can clump together or form interlaced chains which may decrease staining [24]. The tailing phenomena observed in each quadrant can be due to the presence of subpopulations with different levels of resistance to the antimicrobial agent.

Figure 1B shows the percentage of cell viability as a function of the free chlorine concentration used. As it was expected, as the applied concentration of free chlorine increases, the percentage of viable cells decreases, the percentage of dead cells increases and an intermediate sate of damaged cells were observed. To a free chlorine concentration of 5 mg·L^−1^ of 12.7% of the cells remain alive, whereas 66.6% were dead. The other cells are compromised. A higher % of dead cells (82.2%) were obtained to a free chlorine concentration of 50 mg·L^−1^ and a lower % of viable cells were obtained (5.12%). For this concentration, there is still a significant amount of compromised cells. To 100 mg·L^−1^, almost all cells are dead (99.03%).

The classical plate count method only enumerates those cells that can replicate under the specific experimental conditions (CFU), while the premise is that all the others are dead. Thus, quantification of VBNC cells is not possible with this technique because although they still presented metabolic activity, those cells are stressed and lost their ability to grow on agar medium. Moreover, agar plate counting is a time-consuming, labor-intensive, and multi-day process. Nevertheless, culturable cells, obtained with this method (Figure 2), were also determined to monitor the possible existence of VBNC cells of *E. coli* when subjected to different concentrations of sodium hypochlorite. Considering culturable cells (CFU·mL^−1^), there were significant differences between the concentrations of 5 and 10 mg·L^−1^, 5 and 25 mg·L^−1^, 5 and 50 mg·L^−1^, and 5 and 100 mg·L^−1^ (*p* < 0.001), and between concentrations of 10 and 25 mg·L^−1^, 10 and 50 mg·L^−1^, and 10 and 100 mg·L^−1^ (*p* < 0.05).

There was a clear divergence between the traditional technique of plate counts and flow cytometric data. The number of cultivable cells was lower than the number of viable cells. Corroborating other studies, this can be due to cell aggregation after the treatment with disinfectants. According to Hayouni et al. [25], what appears to be a single colony on the plate may be a cell aggregate with more than one culturable cell. Furthermore, this difference points out the presence of stressed subpopulations, which are not able to grow on agar plates [16]. In turn, Massicotte et al. [26] explained this discrepancy, mainly, with the chemical interaction/interference of hypochlorite ions and cell components. According to these authors, chlorine can react with nucleic acids causing a reduction of fluorescence. Cells usually are permeable to PI but are inhibited by bleach rendering them impermeable, leading to false interpretations. Specifically, HOCl ions damage DNA and RNA as well as polynucleotides to form chloramines [27]. Phe et al. [28] also observed an alteration in fluorescence of *E. coli* cells exposed to sodium hypochlorite at a concentration >3 µmol·L^−1^ of chlorine. The difference between fluorescing cells (viable + compromised), obtained by flow cytometry, and culturable cells, obtained by CFU assay, point out to the presence of possible VBNC cells [29] (Figure 3).

There is no formation of VBNC cells only to a concentration of 100 mg·L^−1^ of free chlorine (Figure 3) though this will probably be possible with an intermediate concentration between 50 and 100 mg·L^−1^. Up to this concentration, there was always a high number of VBNC cells. It has been reported that while chlorine has a broad spectrum of antimicrobial activity, its efficacy against pathogens attached to the leafy greens is limited. As chlorine reacts with organic matter, components released from cuts on leaves may neutralize some of the chlorine before it reaches microbial cells [7]. Additionally, crevices, cracks, and small fissures in produce, along with the hydrophobic nature of the waxy cuticle on the surface of many fruit and vegetables, may prevent chlorine and other sanitizers from reaching the microorganisms [30]. In the present study, at a concentration of 10 mg·L^−1^, there was already a reduction in viable cells of almost 3 log, approximately 5 log at 25 mg·L^−1^, and to a concentration higher than 25 mg·L^−1^ almost 7 log reduction. However, according to Van Haute et al. [31], the efficiency of chlorine as a sanitizer for lettuce decontamination is generally limited to 1- to 2-log reductions, even at high chlorine concentrations, being the standard applied free chlorine concentrations in washing processes in the range of 50 to 200 mg·L^−1^. Li et al [32] observed that the survival of *E. coli* O157:H7 on cut lettuce pieces after submersion for 90 s in a solution of 20 mg·L^−1^ chlorine at 20 or 50 °C was not significantly different from the non-chlorine treatment. Beuchat [33] also observed that spraying lettuce with the same concentration of chlorine was as efficient as the treatment with deionized water. A reduction of <1 log CFU·g^−1^ after a 5 min dip in 100 mg·L^−1^ free chlorine compared to a plain water dip when inoculated lettuce leaves with *E. coli* was obtained by Behrsing et al. [34]. Pan and Nakano [35] also observed a reduction of approximately 2 log CFU/g in *E. coli* O157:H7 levels on leafy vegetables, as lettuce was washed with NaClO at 0, 50, 100, 200, and 500 ppm for 5 min compared with the unwashed lettuce. Seo and Frank [36] showed that while *E. coli* O157:H7 cells on the surface of iceberg lettuce leaves were killed by a 20 mg·L^−1^ chlorine decontamination treatment, the cells inside the stomata or those which had penetrated the cut edge of a leaf survived. Niemira [37] reported a reduction of less than 1 log of *E. coli* O157:H7 internalized in 4 lettuce varieties (iceberg, Boston, green leaf, or red leaf lettuce) were obtained with sodium hypochlorite solutions of 300 mg·L^−1^ and 600 mg·L^−1^. This is not according to the results of this study. In fact, for a concentration of 100 mg·L^−1^ of free chlorine all the cells are dead. Probably, the internalization of *E. coli* O157:H7 in the 4 lettuce varieties is responsible for the low efficiency of sodium hypochlorite. In this present study, the cells were not yet internalized.

The potential safety hazards such as the formation of disinfection by-products due to the use of chlorinated compounds pose the need for alternative disinfection systems. The peracetic acid (also known as peroxyacetic acid or PAA, C_2_H_4_O_3_) forms a mixture of hydrogen peroxide (H_2_O_2_) and acetic acid (CH_3_COOH) dissolved in an aqueous solution, which acts as a strong oxidizing agent that is more effective than chlorine or chlorine dioxide. It has been proposed as a sanitizing agent for its usefulness in the fresh-cut industry [38,39]. Besides, it is virtually unaffected by changes in temperature and activity biocide, even in the presence of organic matter [39]. Figure 4A presents representative data for *E. coli* showing dot plots labeled with SYTO-BC and PI treated with different concentrations of PAA. For a concentration of 5 mg·L^−1^ of PAA 64.6% of the cells remain alive, whereas 23.74% were dead (Figure 4B). The other cells are compromised. For a concentration, as much as 50 mg·L^−1^ a percentage of 96.43% are dead cells. Concerning viable cells, there are significant differences between the concentrations of 5, 10, 25, and 50 mg·L^−1^ (*p* < 0.0001). It is also evident that, unlike the previous disinfectant (Figure 3), the relative percentage of viable cells is very similar up to the concentration of 25 mg·L^−1^, with a sharp decrease to 50 mg·L^−1^. Regarding dead cells, there is always an increase. This means that at half the antimicrobial concentration approximately the same % of dead cells are achieved indicating that PAA is a better sanitizing agent.

These results are promising since concentrations below the highest allowable concentration of PAA (80 mg·L^−1^) for fruit and vegetables are efficient to eliminate *E. coli* from lettuce. Moreover, this agent presented a better disinfectant activity than sodium hypochlorite. To the same concentrations applied, the percentage of dead cells is higher and the percentage of VBNC is lower. Using a sanitizer that contained peracetic acid at 40 and 80 mg·L^−1^, Park and Beuchat [40] significantly (*p* < 0.05) reduced *Salmonella* and *E. coli* O157H7 populations on cantaloupe and honeydew melon surfaces. Figure 5 represents the values of culturable cells (obtained by CFU counts) of *E. coli* when subjected to different concentrations of peracetic acid.

It can be seen that the number of CFU·mL^−1^ is higher than those obtained with sodium hypochlorite. This was expected since the number of compromised cells is higher. There were only significant differences between the concentrations of 5 and 25 mg·L^−1^, 5 and 50 mg·L^−1^, 10 and 25 mg·L^−1^, and 10 and 50 mg·L^−1^ (*p* < 0.05). The possible VBNC state of *E. coli* after disinfection with peracetic acid can be seen in Figure 6. 

As it was found for live (viable) and compromised cells, VBNC cells have only significant differences between concentrations of 5, 10, 25 mg·L^−1^, and 50 mg·L^−1^. These results are similar to those obtained by Keeratipibul et al. [41] who investigated the efficacy of hypochlorous and peracetic acids in reducing *E. coli* levels on lettuce. Using a concentration of 50 mg·L^−1^ of peracetic acid, these authors also obtained a reduction of the level of *E. coli* on the lettuce of about 2.5 log CFU·g^−1^. The higher disinfectant activity of PAA may be attributed to its higher oxidation potential compared with chlorine [42].

Viola et al. [43] evaluated the cytotoxicity and the mechanism of cell aggression of peracetic acid in comparison with sodium hypochlorite to L929 cells, through flow cytometry, and showed that both PAA and NaOCl induced cell death predominantly by necrosis instead of apoptosis. According to the authors, this may be explained by the fact that NaOCl and PAA are strong oxidant agents [44], which cause an accentuated damage to the membranes and, consequently, the enzymes extravasate the lysosomes and digest the cell, resulting in necrosis. In the present study, the effect of both disinfectants in *E. coli* is, probably, the same. In fact, through FCM it was possible to verify a lack of integrity of the membrane and the consequent entrance of the fluorochromes. 

## 3. Materials and Methods

### 3.1. Bacterial Strain and Inoculum Preparation

*Escherichia coli* CECT 434 was used in this study. The bacteria cultures were prepared in beads and kept in vials in a freezer at −70 °C. The stock cultures were re-activated by inoculation onto Tryptic Soya Agar (TSA) (Oxoid, UK) plates, which were incubated without shaking for 24 ± 2 h at 37 °C. To prepare the inoculum, a single colony was picked up from stock and incubated in Tryptic Soy Broth (TSB) (Scharlau, Spain) at 37 °C for 24 ± 2 h in a shaking incubator. A subculture was also prepared in TSB at the same temperature and for 18 ± 2 h allowing the bacteria to reach the stationary phase (approximately 10^8^ CFU·mL^−1^).

### 3.2. Lettuce Preparation

Green leaf lettuce (*Lactuca sativa L.*) was purchased from a local producer and was kept in refrigeration conditions at 4 °C. For the experiments, damaged outer leaves of the lettuce were removed. Medium and inner leaves were first washed with running tap water to eliminate any presence of soil or other material, and then with sterile deionized water. It was then dried gently with absorbing paper to drain the water excess. Lettuce leaves were cut using a scalpel into pieces of 2 × 2 cm^2^ and placed under a UV lamp (λ = 253.7) of a flow laminar cabinet (Telstar Bio II Advance) for 15 min on each side to eliminate lettuce’s natural flora (time was selected based on preliminary tests).

### 3.3. Inoculation of Lettuce

Four sterile lettuce samples were transferred to Falcon tubes and submerged into 10 mL of an *E. coli* suspension (approximately 10^8^ CFU·mL^−1^) for 60 min at room temperature [45]. After this period, the cell suspension was discarded and the lettuce samples were immediately washed twice for 1 min with sterile deionized water to remove the bacteria that had not adhered to the lettuce [45,46]. The lettuce samples were then immersed in 30 mL of the intended concentration of sodium hypochlorite or peracetic acid for 5 min. After this period, the solution was decanted, the samples were suspended in 10 mL of a sterile-filtered solution of 0.9% NaCl, and vigorously stirred for 5 s, repeating this procedure 5 times. Finally, 2 mL of each sample was removed to be analyzed by flow cytometry and 200 µL to determine colony forming units (CFU). For each disinfectant concentration, four samples were tested and at least three independent assays were performed.

### 3.4. Washing Solutions

The disinfectants used were 1.5% sodium hypochlorite (HS) with concentrations of free chlorine of 5, 10, 25, 50, and 100 mg·L^−1^ and 35% peracetic acid (PA) (Merck, Darmstadt, Germany), at concentrations of 5, 10, 25, and 50 mg·L^−1^. Dilutions were done in sterile and filtered water. After being prepared they were kept in the dark so they were not altered by light.

### 3.5. Bacterial Viability Assays

#### 3.5.1. Flow Cytometry Analysis

Cell viability was determined by flow cytometry (EC800 Sony Biotechnology Flow Cytometer) using the known Live/Dead to detect the true percentage of dead, live, and compromised *E. coli* CECT 434 cells in lettuce trials after disinfection. The SYTO-BC 5 µM (Thermo Fisher Scientific) and the Propidium iodide (Thermo Fisher Scientific) at a concentration of 20 μg·mL^−1^ were used to evaluate the integrity of the membrane and were excited by the 488 nm laser of the flow cytometer. Live bacteria emitted green fluorescence detected with 525/50 BP on the FL1 channel whilst compromised of dead bacteria emitted yellow to red fluorescence detected with 615/30BP on the FL4 channel, respectively. Samples were incubated in the dark for 10 min. During acquisition, all parameters were collected in log mode and data analysis was carried out with the EC800 software version 1.3.6. (Sony Biotechnology Inc., Champaign, IL, USA). Forward and Side Scatter gates were established to exclude debris. Unstained control, live untreated culture, and isopropanol 70% dead culture were used for gating the different regions. Single stained and double-stained cultures were used for compensation purposes. In addition, to optimize the protocol and to reduce noise (particles that are not considered events), the disinfectants were prepared with sterile filtered water and the samples were finally suspended in 10 mL of a sterile-filtered solution of 0.9 % NaCl.

#### 3.5.2. Standard Plate Count Method

Colony-forming units (CFU) were obtained as follows: a 200 µL aliquot was removed from each assay flask. Serial 10-fold dilutions were made in saline solution and plated in TSA. Colonies were counted after 24 h of incubation at 37 °C. Final data, given as log CFU·mL^−1^, resulted from at least three independent experiments with three replicates each.

### 3.6. Statistical Analysis

Statistical analysis was done using GraphPad Prism 6. Data are presented as mean ± standard deviations (SD) based on triplicates from at least three independent experiments. Data were compared using two-way ANOVA, with Tukey’s multiple comparison statistical test. *p* ≤ 0.05 was considered statistically significant (95% confidence interval).

## 4. Conclusions

It is well known that, during processing, excessive concentrations of antimicrobial chemicals are truly inadvisable as they can damage equipment, reduce product quality, be harmful to worker health, and may pose a hazard to consumers. In addition, for economic reasons, these concentrations should be as low as possible. With this study, it can be concluded that the recommended levels of chemical disinfectants for fresh fruit and vegetables are adequate. More importantly, it is possible to ensure that all *E. coli* cells are dead and that no VBNC cells are present with lower concentrations of these chemicals. It was also possible to demonstrate that flow cytometry (FCM) is a powerful tool for rapidly evaluating the effect of disinfection on *E. coli* viability in lettuce as with FCM, besides live and dead cells, intermediate states as viable but nonculturable cells are also detected. Therefore, the detection of VBNC cells is possible within a few minutes, which avoids their persistence, spread and consequently, foodborne outbreaks. This demonstrates that this technique can be advantageous in the field of food safety and of medical microbiology.

## Figures and Tables

**Figure 1 antibiotics-09-00014-f001:**
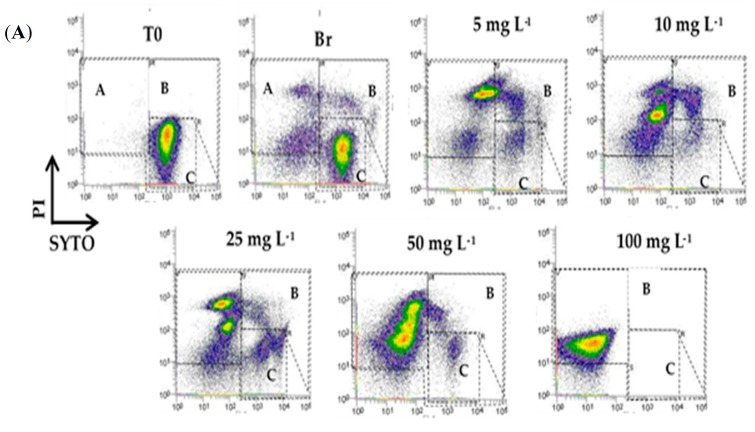
Effect of different concentrations of free chlorine in the physiological state of *Escherichia coli* CECT (Colección Española de Cultivos Tipo) 434 cells. (**A**) Flow cytometry (FCM) representative data for *E. coli* showing dot plots labeled with SYTO-BC and PI. Regions are sorted by A: dead cells, B: compromised cells, and C: viable cells. The figures are ordered by T0: initial concentration of inoculated cell suspension in the lettuce samples, B: the negative control in which there was no disinfectant used, and wash was performed with sterile filtered water, and the remaining figures refer to the different concentrations of free chlorine. The events in the overlay histograms were normalized. (**B**) Percentage of live, dead, and compromised cells concerning total cell numbers (100%) after each treatment. These data were collected from three independent assays and made in triplicates. The values presented correspond to the means and respective standard deviations of the nine replicates, independently analyzed.

**Figure 2 antibiotics-09-00014-f002:**
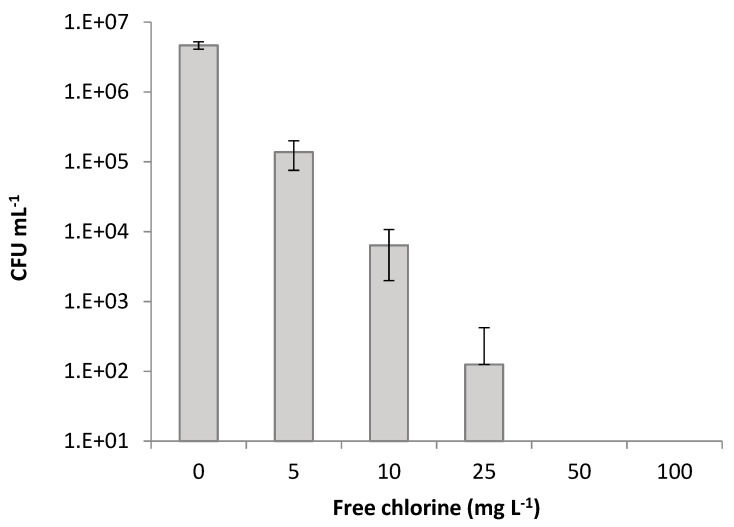
Cell culturability of *Escherichia coli* CECT 434 after the treatment with different concentrations of free chlorine. Data were collected from three independent trials, elaborated in triplicates. Results correspond to the means and respective standard deviations of the nine replicates.

**Figure 3 antibiotics-09-00014-f003:**
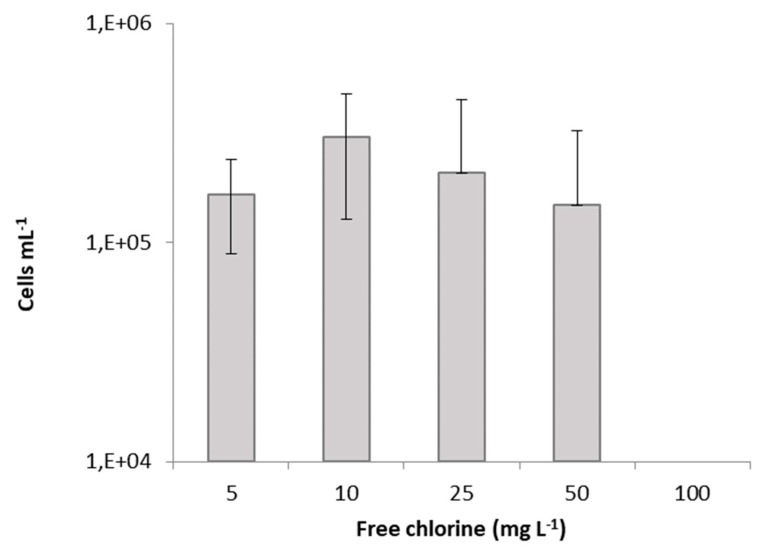
Induction of viable but nonculturable bacteria (VBNC) of *Escherichia coli* CECT 434 after treatment with different concentrations of free chlorine. Data were collected from three independent trials, elaborated in triplicates. Results correspond to the means and respective standard deviations of the nine replicates.

**Figure 4 antibiotics-09-00014-f004:**
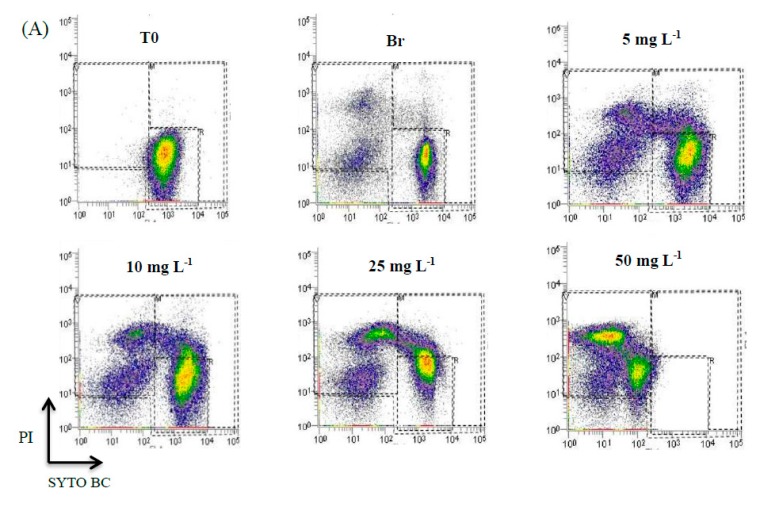
Effect of different concentrations of peracetic acid in the physiological state of *Escherichia coli* CECT 434 cells. (**A**) FCM representative data for *E. coli* showing dot plots labeled with SYTO BC and PI. Regions are sorted by A: dead cells, B: compromised cells, and C: viable cells. The figures are ordered by T0: initial concentration of inoculated cell suspension in the lettuce samples, Br: the negative control in which there was no wash with disinfectant, but with sterile filtered water, and the remaining figures refer to the different concentrations of peracetic acid. The events in the overlay histograms were normalized. (**B**) Levels of live, dead, and compromised cells in relation to total cell number (100%) after each treatment. These data were collected from three independent assays and made in triplicates. The values presented correspond to the means and respective standard deviations of the nine replicates, independently analyzed.

**Figure 5 antibiotics-09-00014-f005:**
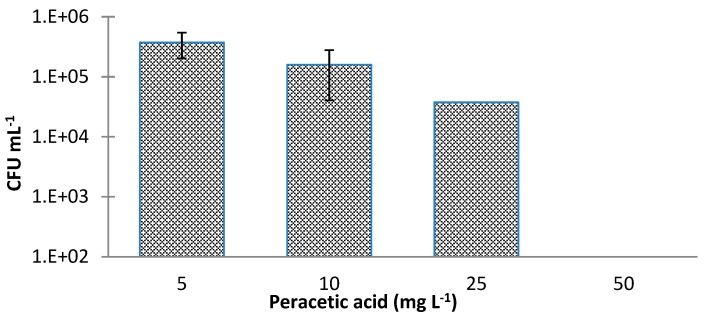
Cell culturability of *Escherichia coli* CECT 434 after treatment with different concentrations of peracetic acid. Data were collected from three trials, performed on different independent days, and elaborated in triplicates. Results correspond to the means and respective standard deviations of the nine replicates.

**Figure 6 antibiotics-09-00014-f006:**
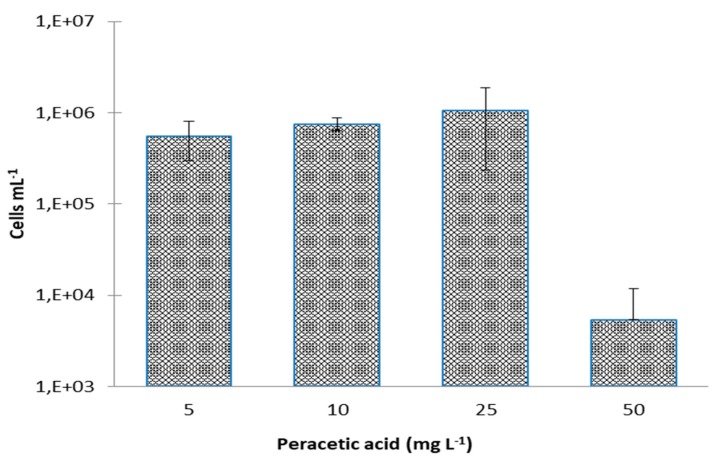
Induction of VBNC of *Escherichia coli* CECT 434 after treatment with different concentrations of peracetic acid. Data were collected from three independent trials, elaborated in triplicates. Results correspond to the means and respective standard deviations of the nine replicates.

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
