# Peer review of "Evaluation by Flow Cytometry of Escherichia coli Viability in Lettuce after Disinfection"

_antibiotics, 2019, doi:10.3390/antibiotics9010014_

Round 1

Reviewer 1 Report

REVIEW REPORT

ARTICLE TITLE: Evaluation by flow cytometry of Escherichia coli viability in lettuce after disinfection

ORIGINALITY: Aspects of the work can be considered as contribution to knowledge about disinfection strategies to reduce VBNC bacteria in leafy greens.

SIGNIFICANCE: Significant

QUALITY OF PRESENTATION: Conclusions are linked to main theme of the research.

SCIENTIFIC SOUNDNESS: Methods employed are adequate.

INTEREST TO READERS: Appropriate.

OVERALL MERIT: Average

ENGLISH LEVEL: Needs improvement

OVERALL RECOMMENDATIONS: Accept after minor revisions.

SUMMARY

The paper is aimed at evaluating the efficacy of two disinfectants (Peracetic acid and sodium hypochlorite) in reducing the viability of E. coli in lettuce and examining physiological states of the cells using flow cytometry. The research indicates that peracetic acid is more effective than sodium hypochlorite, inducing lower percentages of VBNC and higher levels of dead cells.

BROAD COMMENTS

Grammar should be corrected throughout the text as author made several grammatical errors including typos, wrong phrases/tenses.

SPECIFIC COMMENTS

ABTRACT

Line 10-16: Introduction in the abstract should be limited to two brief sentences.

Line 16-17: Specific methodology utilized for the assay should be briefly mentioned in the abstract

Line 21-23: the concentrations of disinfectants used and the effective levels should also be stated in the abstract.

Keyword; leafy greens should be replaced with lettuce.

INTRODUCTION

Line 30-31: Sentence should be rephrased

Line 33, 37 and 46: sentence should be corrected

Line 47-50. Citation needed

Line 55-59: the mechanism of action should be included

Line 60-61: rephrase

Line 74: rephrase

Line 80-84: include how VNBC can be differentiated from living cells

RESULTS AND DISCUSSIONS

Line 90: insert full definition of Pi (Propidium iodide) as it is the first time it is being mentioned.

Line 102-110: move to introduction section

Line 122. Figure should be labelled as Figure 1A and description of figure inserted underneath

Line 123: Figure should be labelled as Figure 1B and description of figure inserted underneath

Line 134: Start statement by describing the figure. Eg. ‘Figure 1B shows percentage…….’

Line 138 and 139: correct grammar

Line 148: state the colony forming units

Line 155: remove “as can be seen”

Line 158: remove ‘in fact, and’

Line 159 and 160: replace ‘besides and more important’ with FURTHERMORE

Line 167: remove ‘besides, it has also been hypothesized the possible………………………..’

Line 155-170: link this section discussion to why there is a difference between results of flow cytometry assay and culture method, explaining why this is due to VBNC

Line 175: Rephrase

Line 177-180: Citation needed

Line 183-185: Rephrase

Line 201-204: Did the cutting, agitation and rinsing steps in sample preparation affect this reduction in bacterial load as compared to previous studies cited?

Line 211-213: Present result first before making the statement.

Line 218-219: Rephrase

Line 222: add ‘indicating that peracetic acid is a better sanitizing agent.

Line 224. Figure should be labelled as Figure 4A and description of figure inserted underneath

Line 225. Figure should be labelled as Figure 4B and description of figure inserted underneath

Line 236: restate the results first

Line 238: correct grammar

Line 247-248: justify why

Line 262: correct to ‘Viola et al.,

Line 266-267: rephrase

MATERIALS AND METHODS

Line 275: was the incubation with or without shaking?

Line 276: correct grammar

Line 289: was the weight of the samples measured?

Line 290-292: was the washing done immediately? Clarify.

Line 293: state the concentrations of disinfectants ued

Line 294: was there a rinse step? Clarify.

How many replicates were done?

Line 329: correct spelling

CONCLUSIONS

Line 334-337: rewrite

Line 339-341: rewrite

Author Response

We would like to thank you for the revision concerning the manuscript. All the comments and suggestions were taken into consideration and have surely contributed to its improvement and we truly hope that this revision is in accordance with the reviewer’s indications. All modifications to the initial manuscript are indicated in green in the revised manuscript. Furthermore, the answers to the reviewer comments are presented next. We would like to reinforce that the English was carefully revised, the grammar was corrected as well as the specifics remarks.

INTRODUCTION

Line 55-59: the mechanism of action should be included

We think that the mechanism of action is already explained in the text.

RESULTS AND DISCUSSIONS

Line 155-170: link this section discussion to why there is a difference between results of flow cytometry assay and culture method, explaining why this is due to VBNC

On lines 146-150 it is explained, in general, why there is a difference between the results of flow cytometry assay and culture method and on lines 155-170 we try to apply this justification, as well as others, to our own results.

Line 201-204: Did the cutting, agitation and rinsing steps in sample preparation affect this reduction in bacterial load as compared to previous studies cited?

No, we don’t think so. We think that this happens because “our cells” are not yet internalized.

Line 236: restate the results

This sentence refers to the results shown in the previous figure and described just before the figure.

Line 266-267: rephrase

MATERIALS AND METHODS

Line 289: was the weight of the samples measured?

No, the samples were just measured, the leaves were cut in 2x2 cm2 squares.

Line 290-292: was the washing done immediately? Clarify.

Yes, immediately and for one minute.

Line 293: state the concentrations of disinfectants used

The concentrations of disinfectants are indicated immediately in the next section, section 3.4.

Line 294: was there a rinse step? Clarify.

No, there wasn’t.

Reviewer 2 Report

General assessment and major comments:

In the submitted study, Teixeira et al. combined flow cytometry (FCM) based live/dead cell analysis and traditional plate counting method to study the induction of VBNC (in E. coli) post exposure to disinfectants. The authors found that viability profile of bacteria obtained from flow cytometry analysis are different from plate based analysis. Based on this observation, the authors deduced the presence of VBNC and calculated the number. Using FCM based analysis to investigate the status and viability of bacteria post disinfectants exposure is interesting and important, which will provide helpful information to the readers in the field who are interested in this topic. 

Here are the suggestions for authors to consider.

1. The authors did not mentioned how the number of VBNC is calculated.

2. The labels in figures 1 and Figure 4 are hard to read, some of the labels are masked by the figures.

3. Line 134 , line 135, and line 222, change “%” to “percentage”. 

Author Response

We would like to thank you for the revision concerning the manuscript. All the comments and suggestions were taken into consideration and have surely contributed to its improvement and we truly hope that this revision is in accordance with the reviewer’s indications. All modifications to the initial manuscript are indicated in green in the revised manuscript. Furthermore, the answers to the reviewer comments are presented next. We would like to reinforce that the English was carefully revised, the grammar was corrected as well as the specifics remarks.

The authors did not mentioned how the number of VBNC is calculated.

 In lines 168-170 it is explained that “The difference between fluorescing cells (viable + compromised), obtained by flow cytometry, and culturable cells, obtained by CFU assay, point out to the presence of possible VBNC cells [29]”.

Reviewer 3 Report

The manuscript on evaluation of E.coli viability by flow cytometry after the use of commonly used chemical disinfectants on lettuce has covered interesting aspects. Author have represented how they have used flow cytometry  to  access the actual microbial load i.e VBNC and culturable E.coli in lettuce after treating it with different concentrations of disinfectants.

Comments:

1)LINE: 11-13; 66-71;82-84 need to be restructured.

2)It will be good if the author can condense the discussion parts in result and discussion section and emphasize a bit more on their result area. 

Author Response

We would like to thank you for the revision concerning the manuscript. All the comments and suggestions were taken into consideration and have surely contributed to its improvement and we truly hope that this revision is in accordance with the reviewer’s indications. All modifications to the initial manuscript are indicated in green in the revised manuscript. Furthermore, the answers to the reviewer comments are presented next. We would like to reinforce that the English was carefully revised, the grammar was corrected as well as the specifics remarks.

2)It will be good if the author can condense the discussion parts in result and discussion section and emphasize a bit more on their result area. 

We think the results are already condensed with the discussion and it is not easy to do so more clearly. We've moved a few sentences from one place to another and hope the reviewer thinks this section is better.

Round 2

Reviewer 2 Report

In the revised manuscript, the authors have significantly improved the quality of presentation. 

There are still some minor points in figures need to be corrected.

for instance, in figure 1A, it should be "PI" instead of "P".

figure 4A, all the numbers of concentration are masked by the flow cytometry images.